# PolyAudio: Advancing Multi-Audio Reasoning in Large Audio Language Models

## Abstract

While Large Audio Language Models (LALMs) have achieved superior performance on reasoning over single audio clips, their ability to understand and reason over *multiple audio clips* remains a significant challenge. In this paper, we introduce PolyAudio, a novel LALM specifically designed for this complex task. To systematically train and evaluate our model, we first identify and formalize eleven foundational multi-audio reasoning capabilities. These capabilities, spanning sound, music, and speech, are designed to represent a broad range of challenging real-world scenarios. To enhance these skills, we fine-tune the Qwen2-Audio-7B-Instruct model using Group Relative Policy Optimization (GRPO). This approach mitigates common issues associated with Supervised Fine-Tuning (SFT), such as catastrophic forgetting. Specifically, we construct preference data that explicitly rewards the model for correctly synthesizing information across multiple audio clips. Our model, PolyAudio, achieves **58.6%** on the MMAU-Pro multi-audio subset and **71.2%** on our PolyAudio-Bench, substantially outperforming baselines on multi-audio reasoning tasks while maintaining its performance on single-audio tasks. To promote research in this space, we will publicly release the model, data generators, evaluation scripts, and training recipes at the time of publication.

## 1 Introduction

Recent state-of-the-art LALMs and multimodal models like Gemini-2.5 (et al, 2025), Qwen-2.5-Omni (Xu et al., 2025), and Audio Flamingo 3 (Goel et al., 2025) have demonstrated superior audio understanding capabilities well beyond the scope of traditional tasks like Automatic Speech Recognition (ASR). These models are trained on audio-text question-answer pairs, leveraging natural language as a more descriptive supervision signal compared to predefined labels for complex, real-world audio recordings. This has led to impressive performance on a variety of single-audio tasks, such as spoken language understanding (Tang et al., 2023; Gong et al., 2023), audio captioning (Ghosh et al., 2024a; Deshmukh et al., 2023), and complex reasoning (Ghosh et al., 2024b; 2025; Goel et al., 2025; Xie et al., 2025). For example, natural language can describe the temporal order of multiple events (Ghosh et al., 2023) using words such as 'simultaneous', 'before', and 'after', which helps models learn the intrinsic relationships within an audio stream more effectively. This success, however, primarily applies to single audio inputs, where the model processes a self-contained unit of information and applies its powerful language reasoning capabilities to generate a response.

Despite significant progress in audio understanding and reasoning, the capacity of LALMs for compositional and contextual understanding and reasoning across multiple discrete audio clips remains a research challenge. While existing models can achieve near-human performance on isolated tasks, their ability to synthesize information from a collection of different clips, such as identifying a common speaker, establishing a chronological sequence, or verifying factual consistency, is a fundamentally different problem. Tackling this requires models with new capabilities for inter-clip reasoning, along with novel training paradigms and evaluation benchmarks to drive progress.

The research direction in multi-modal visual understanding (?Ye et al., 2024; Zhu et al., 2025) offers a compelling precedent for the study of multi-audio reasoning. Early Large Vision-Language Models (LVLMs) achieved considerable performance on single-image tasks, but this initial focus soon gave way to a broader recognition that real-world scenario requires reasoning across multiple images. This steered the research direction to the creation of specialized benchmarks designed

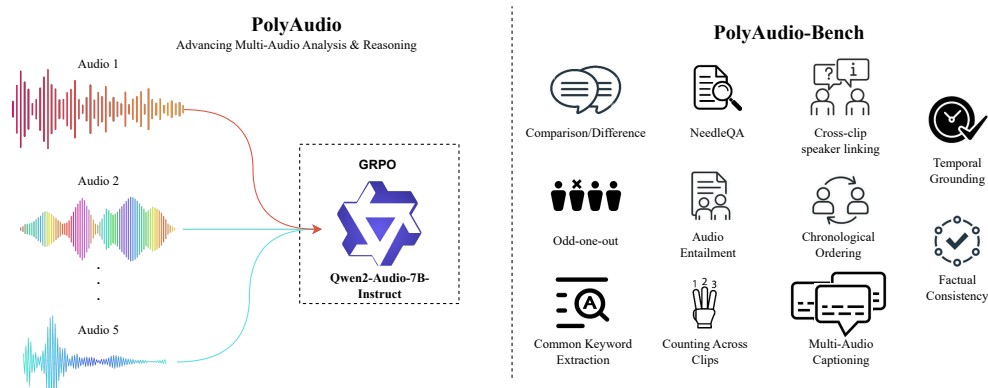

Figure 1: Overview of POLYAUDIO & POLYAUDIO-BENCH. POLYAUDIO leverages GRPO for enabling multi-audio analysis & reasoning. POLYAUDIO-BENCH comprises 11 task categories to train and benchmarks understanding and reasoning over multiple audio clips up to 5 clips.

specifically to evaluate these more challenging, multi-instance reasoning abilities. To address this, benchmarks such as the MMIU (Meng et al., 2024) and MIRB (Zhao et al., 2024) were introduced. These evaluations focus on multi-image relationships spanning semantic, temporal, and spatial reasoning, and have revealed significant challenges that persist even for state-of-the-art models. For instance, the modest 55.7% accuracy of GPT-4o (OpenAI et al., 2024) on MMIU underscores that multi-instance reasoning is a fundamental, multi-modal challenge, not a problem confined to a single modality.

## 1.1 OUR CONTRIBUTION

In this paper, we propose POLYAUDIO, a novel training and evaluation approach to pioneer multi-audio understanding and reasoning. We design our dataset and training pipeline to probe the relational and compositional reasoning capabilities that are fundamental to genuine multi-audio understanding and reasoning. Our contributions are as follows:

- We introduce POLYAUDIO, a novel Large Audio Language Model specifically designed for complex reasoning across multiple (up-to 5), discrete audio clips of up to 30 seconds each, bridging a critical gap in current LALM capabilities.

- We introduce two curated datasets to enable multi-audio reasoning: POLYAUDIO-TRAIN containing 110,000 question-answer pairs, and POLYAUDIO-PREF a preference dataset with 50,000 examples. Both datasets consist of up to five audio clips per QA pair..

- We also propose the POLYAUDIO-BENCH a comprehensive suite of eleven distinct tasks and corresponding data generators designed to systematically evaluate multi-audio reasoning. The benchmarks consists of 2750 question-answer pairs, i.e, 250 QA pairs for each skill.

- We propose using Group Relative Policy Optimization (GRPO) on our POLYAUDIO-PREF to mitigate the adverse effects of SFT like catastrophic forgetting (Rajani et al., 2025), specifically aligning the model for correctness, reasoning over multiple audio clips, and consistency.

- We demonstrate that POLYAUDIO substantially outperforms strong concatenation-based and other existing baselines on our benchmark, particularly on challenging tasks like difference detection, factual consistency, and counting, while maintaining competitive performance on single-audio tasks.

| Method/Benchmark | Primary Focus | Max. #Audio | Task Scope | Key Contribution |
|---|---|---|---|---|
| ADIFF | Paired-Audio Difference Explanation | 2 | 1 Task (Difference Captioning) | Cross-Projection Layer for contrastive learning |
| Mellow | Paired-Audio Comparative Reasoning | 2 | ∼7 Reasoning Tasks (ReasonAQA) | Optimized for small model size |
| MALLM / MAE | General Multi-Audio Processing | 2 | 11 Simplified Tasks (MAE Benchmark) | First model/benchmark for multi-audio tasks |
| AIR-Bench | Generative Comprehension | 1 | 19 Foundation + Chat Tasks | Audio mixing strategy for complex single clips |
| MMAU | Single-Audio Expert Reasoning | 1 | 27 Expert-level Skills | Benchmark for deep, domain-specific knowledge |
| MMAU-Pro | Single and Multi-Audio Reasoning | 3 | 49 Expert Skills | Benchmark for complex audio reasoning over multiple domains |
| POLYAUDIO/ POLYAUDIO-BENCH | N-Way Multi-Audio Reasoning | 5 | 11 Task Categories | Multi-audio analysis & Reasoning |

Table 1: Comparison of POLYAUDIO with existing LALMs and benchmarks. We highlight that while prior work has predominantly focused on reasoning over single or paired audio inputs, POLYAUDIO is uniquely designed for complex N-way reasoning across multiple (2-5) audio clips.

## 2 RELATED WORK

### 2.1 LARGE AUDIO LANGUAGE MODELS & FOUNDATION MODELS

Large Audio Language Models (LALMs) are a specific class of LMMs focused on audio understanding and generation. They move beyond the traditional paradigms of speech recognition by using natural language supervision, where models learn from descriptive text rather than predefined labels. This approach allows LALMs to understand and reason about complex audio events, including spoken language, natural sounds, and music, enabling applications such as audio captioning, music composition, and virtual assistants. While some LALMs like Mellow (Deshmukh et al., 2025) and MALLM (Chen et al., 2024) try to perform multi-audio analysis on up to two audio clips, the majority of LALMs are still restricted to single audio analysis, and research efforts have focused on improving performance on single, contiguous audio streams. Our work directly targets this limitation by introducing a model purpose-built for multi-clip reasoning.

### 2.2 REASONING ACROSS MULTIPLE AUDIO CLIPS

Recent work has begun to explore reasoning beyond a single audio clip, primarily focusing on pairwise comparisons. Deshmukh et al. (2025) introduced ADIFF, the first comprehensive approach for explaining differences between two audio recordings in natural language. The ADIFF model uses a cross-projection layer to facilitate contrastive analysis between two audio inputs, but this architectural choice inherently limits its application to pairs of clips. This N=2 paradigm is also present in other models like Mellow, a compact LALM designed for comparative reasoning tasks that also takes two audio clips as input.

The first work to explicitly address a "multi-audio era" was introduced by Chen et al. (2024) with the MALLM model and the Multi-Audio Evaluation (MAE) benchmark. The MAE benchmark includes 11 multi-audio tasks, and the MALLM model is trained using a discriminative learning approach on synthetically generated audio pairs to capture inter-clip context. However, a closer examination reveals that the MAE benchmark is constructed from samples that each represent a combination of only two audio contexts, and the training strategy is centered on an "audio pairs synthesis" approach. Consequently, despite its framing, this work is fundamentally limited to pairwise reasoning and does not evaluate or enable generalized reasoning across more than two discrete audio clips. Our work addresses this critical gap by introducing a model and a benchmark specifically designed for compositional reasoning across up to five audio clips, evaluated on eleven distinct and complex task categories.

## 2.3 AUDIO EVALUATION BENCHMARKS

The evaluation of LALMs has evolved from assessing fundamental perceptual tasks to probing deeper cognitive abilities. Early benchmarks like SUPERB (Yang et al., 2021) and HEAR (Turian et al., 2022) focused on a suite of classification and detection tasks to measure the generalizability of audio representations. With the rise of instruction-tuned models, benchmarks shifted to evaluating generative comprehension. AIR-Bench, for instance, was the first benchmark designed to assess the instruction-following capabilities of LALMs across speech, sounds, and music. It introduced a novel audio mixing strategy, where two clips are combined into a single stream to increase input complexity, though the evaluation remains on a single audio input (N=1).

Recent benchmarks like MMAU (Sakshi et al., 2025), MMAR (Ma et al., 2025), MMSU (Wang et al., 2025) aim to assess deeper reasoning and understanding capabilities of these LALMs. MMAU contains over 10,000 human-annotated questions that require domain-specific knowledge and complex reasoning across 27 distinct skills. MMAR and MMSU contain 1000 and 5000 question-answer pairs aimed towards assessing complex reasoning over audio. However, their focus remains on the reasoning depth for a single audio clip. These benchmarks highlight a clear research gap: while existing evaluations thoroughly probe reasoning on single audio inputs, they do not address the challenge of compositional reasoning across multiple, discrete audio streams.

Building on this, MMAU-Pro (Kumar et al., 2025) was introduced as a more challenging benchmark for holistic audio intelligence, expanding to 49 unique auditory skills. Crucially, MMAU-Pro is one of the first benchmarks to explicitly include multi-audio understanding, with 430 instances featuring two audio clips and 26 instances with three. It also incorporates other complex dimensions such as long-form audio comprehension (up to 10 minutes) and spatial audio reasoning. Despite these advances, performance on the multi-audio tasks remains a significant challenge, with no model surpassing 30% accuracy, underscoring the difficulty of multi-clip reasoning and the need for specialized model architectures and training paradigms. These benchmarks highlight a clear research gap: while existing evaluations thoroughly probe reasoning on single audio inputs or have just begun to introduce multi-audio scenarios, they do not provide a comprehensive approach for evaluating generalized, compositional reasoning across multiple, discrete audio streams.

## 2.4 TRAINING METHODOLOGIES FOR MODEL ALIGNMENT

The evolution of foundation models is closely tied to advancements in training methodologies, particularly those for aligning models with human preferences. The process often begins with Supervised Fine-Tuning (SFT) on high-quality, human-curated datasets to teach a model to follow instructions. A more sophisticated approach is Reinforcement Learning from Human Feedback (RLHF) (Ouyang et al., 2022), which uses a reward model trained on human preference pairs to guide the policy of the LLM. This method allows models to learn from subjective human judgments and improve their ability to solve complex tasks where the desired output is difficult to define.

A more recent and efficient alternative to RLHF is Direct Preference Optimization (DPO) (Rafailov et al., 2023), which simplifies the process by eliminating the need for a separate reward model. DPO directly optimizes the language model policy using a straightforward cross-entropy loss function based on human preference pairs. While DPO is more stable and computationally efficient than traditional RLHF, the field continues to seek methods that are even more effective at improving model reasoning and are better suited for large-scale training.

This has led to the development of Group Relative Policy Optimization (GRPO) (Shao et al., 2024), an improvement over Proximal Policy Optimization (PPO) (Schulman et al., 2017), which was famously used for models like ChatGPT. GRPO's core innovation lies in its ability to remove the need for a separate value model. Instead of estimating a reward for a single output, GRPO generates a group of different outputs for the same prompt and uses their rewards relative to the group average as a baseline. This group-based comparison reduces the noise and variance in the feedback signal, leading to faster convergence and a significant reduction in memory usage and training time. This approach is particularly well-suited for improving a model's reasoning abilities, as it allows for nuanced alignment with human preferences that prioritize correctness, efficient clip usage, and consistency across overlapping queries. The choice to use GRPO for fine-tuning the Qwen2-Audio-7B-Instruct model reflects a sophisticated technical decision to employ a methodology uniquely aligned with the complex, multi-faceted nature of our proposed multi-audio reasoning tasks.

| Task Name | Description | Example Prompt |
|---|---|---|
| Comparison/Difference | Compares and contrasts two or more clips. | What is the difference between the first and third audio clips provided? |
| Temporal Grounding | Identifies the time interval of an event based on a natural language description. | In which of the clips does the sound of 'a car driving by' occur, and at what timestamp? |
| Needle QA | Finds and answers a specific question hidden within one of the clips. | A key fact is mentioned in one of the clips. What is the name of the speaker who states the fact about the number of birds? |
| Cross-clip Speaker Linking & Attribution | Identifies and links a single speaker across multiple clips. | Audio clip 1 and audio clip 3 contain speech from the same person. Who is it? |
| Odd-one-out | Identifies the clip that is semantically or thematically distinct from the others. | Which of these three audio clips does not belong with the rest? |
| Multi-Audio Entailment | Determines if the information in one set of clips supports or contradicts a claim made in another. | Based on the first two clips, is the statement 'The weather is sunny today' supported by the third clip? |
| Common Keyword Extraction | Extracts a keyword or phrase that is common to a set of clips. | What is the single keyword spoken in all of the audio clips? |
| Chronological Ordering | Arranges a set of clips in their correct temporal sequence. | These four audio clips were recorded in order. Please arrange them chronologically. |
| Multi-audio Captioning (single and aggregated) | Generates a single or aggregated caption describing the content of multiple clips. | Provide a brief summary of the events that occurred across all three audio clips. |
| Factual Consistency | Verifies factual information across multiple clips and identifies any inconsistencies. | Clip 1 states a person's age. Clip 2 states a different age. What is the correct age, and which clips contain the correct information? |
| Counting across clips | Counts the number of occurrences of an object or event across a set of clips. | How many times does a dog bark across all four audio clips? |

Table 2: Multi-audio Analysis & Reasoning tasks in POLYAUDIO.

# 3 POLYAUDIO: ADVANCING MULTI-AUDIO REASONING IN LARGE AUDIO LANGUAGE MODELS

Formally, the task of N-way multi-audio reasoning can be defined as follows: given a set of N discrete audio clips, $A = \{a_1, a_2, \ldots, a_N\}$, and a natural language query $q$, the model $M$ is required to generate a textual response $r$ that accurately answers the query by synthesizing information across the provided clips. The generation process is conditioned on both the query and the full set of audio inputs: $r = M(q, a_1, a_2, \ldots, a_N)$.

To enable the large language model backbone to process these multiple, discrete audio inputs, we adopt an interleaved text-audio representation scheme. Each audio clip $a_i$ is first processed by a dedicated audio encoder, which transforms the raw waveform into a sequence of feature embeddings, $E_i = \text{AudioEncoder}(a_i)$. These embeddings are then represented in the textual input stream by special placeholder tokens, <audio>.

Our model, POLYAUDIO, is built upon the Qwen2-Audio-7B-InstructChu et al. (2024) architecture, which natively supports this interleaved format. The input to the model is constructed by embedding these <audio> tokens within the natural language prompt. For instance, a user query is formatted as a sequence where text and audio placeholders are interspersed. An example prompt for a comparison task involving three audio clips would be structured as follows:

> "<audio> <audio> <audio> What is the difference between the first and third audio clips provided?"

During processing, the Qwen2-Audio-7B-Instruct's processor replaces each <audio> token with the corresponding sequence of pre-computed audio embeddings ($E_1, E_2, E_3$). This creates a unified sequence of text and audio tokens that is fed into the model. This approach allows the model to leverage its powerful language understanding capabilities to establish relationships, compare con-

tent, and synthesize information across the distinct audio clips, treating them as first-class citizens within the input context.

## 3.1 MULTI-AUDIO ANALYSIS & REASONING TASKS

POLYAUDIO is built upon a comprehensive training and evaluation benchmark that evaluates eleven distinct multi-audio reasoning capabilities. These tasks are designed to move beyond simple perception and assess a model's ability to perform relational, compositional, and semantic reasoning over a collection of audio clips. The tasks can be grouped into a hierarchy of increasing complexity.

**Perceptual and Identificational Tasks**, serve as a foundation, requiring fundamental signal processing and simple aggregation. This includes tasks such as Common Keyword Extraction and Counting across clips. These tasks test a model's ability to accurately transcribe and identify core information from multiple inputs before a more complex understanding is required.

**Relational and Comparative Tasks**, require the model to understand how clips relate to each other in terms of content, time, and uniqueness. Tasks in this category include Comparison/Difference, Odd-one-out, Chronological Ordering, and Temporal Grounding. The Temporal Grounding task, for example, extends a concept from video reasoning by requiring the model to localize a specific event across a set of audio clips, a capability that relies on joint comprehension of audio and language.

**Compositional and Semantic Reasoning**, is the most challenging, demanding a deeper, holistic understanding category of the combined audio content. These tasks, such as Needle QA, Multi-Audio Entailment, Cross-clip Speaker Linking & Attribution, Factual Consistency, and Multi-audio Captioning, require the model to synthesize information, resolve ambiguities, and generate coherent responses. Factual Consistency, for instance, requires the model to compare and verify information across multiple audio segments, an extension of the textual concept that measures whether a model prefers factually consistent continuations of its input. Similarly, Cross-clip Speaker Linking & Attribution addresses the challenge of identifying and linking a single speaker across different, non-contiguous audio segments, a task that has been explored in a traditional context but is a novel challenge for LALMs. Table 2 provides a detailed breakdown of the eleven tasks, including a definition and a concrete example for each.

To systematically train and evaluate these capabilities, we introduce a suite of three distinct datasets. For the initial fine-tuning, we developed **POLYAUDIO-TRAIN**, a large-scale dataset of 110,000 question-answer pairs. To align the model for correctness and reasoning, we curated **POLYAUDIO-PREF**, a dataset of 50,000 QA pairs used for GRPO training. Finally, for evaluation, we constructed **POLYAUDIO-BENCH**, a comprehensive benchmark with 2,750 challenging instances designed to rigorously test performance across all eleven task categories.

## 3.2 DATA AND EVALUATION PIPELINES

### 3.2.1 TRAINING DATA

We train on POLYAUDIO-TRAIN and POLYAUDIO-PREF datasets in 2 consecutive stages which comprises audios from (i) open-source corpora-AudioSet (Gemmeke et al., 2017), PicoAudio (Xie et al., 2024), LibriSpeech (Panayotov et al., 2015), MusicCaps (Agostinelli et al., 2023), Music4All (Santana et al., 2020), Clotho (Drossos et al., 2020), and AudioCaps (Kim et al., 2019)—and (ii) synthetic speech generated as follows: GPT-5 produces transcripts, which are rendered with Higgs Audio v2 (Boson AI, 2025). We create question–answer pairs using GPT-5 and GPT-OSS-120B (OpenAI, 2025). The prompts for each task type are listed in Appendix **??**.

### 3.2.2 EVALUATION

We evaluate system outputs with an automated judge, adopting the LLM-as-a-judge paradigm popularized in recent text and multimodal evaluation. Specifically, we use the open-source **Qwen3-7B-Instruct** as our judge for reproducibility and cost efficiency. For each item, the judge receives (i) the question, (ii) the ground truth answer and (iii) the model's answer. A rubric-guided prompt instructs the judge to produce a *scalar score* in $[1, 5]$ based on two criteria: *factual correctness with respect to the provided clips*, and *parsimony of clip usage* (penalizing answers that rely on irrelevant clips). While evaluating LLM-as-a-judge, we average over two paraphrased judge prompts. Inspired by

prior LLM-as-a-judge protocols, we report only the judge's scalar score as our main metric across tasks. To validate this setting, we obtain human scores on 100 items from our POLYAUDIO-BENCH scored under the same rubric and report the correlation between human and judge scores (Spearman's $\rho$ and Kendall's $\tau$) in Table 3.

| LLM | Spearman's $\rho$ | Kendall's $\tau$ |
|---|---|---|
| Qwen-3 7B Instruct | **0.748** | **0.692** |
| Llama 3.1 8B | 0.675 | 0.612 |

Table 3: Correlation between human and LLM-as-a-Judge for evaluating model responses for answer correctness on POLYAUDIO-BENCH.

## 4 EXPERIMENTS

To validate the effectiveness of POLYAUDIO, we conduct a comprehensive set of experiments designed to assess its multi-audio reasoning capabilities. We evaluate our model on our newly proposed POLYAUDIO-BENCH benchmark and the multi-audio subset of MMAU-Pro, comparing its performance against strong baselines. We also test the model on audio understanding on single audio clip to ensure that adding multi-audio training doesn't degrade the performance on general audio understanding tasks.

### 4.1 EXPERIMENTAL SETUP

#### 4.1.1 EVALUATION BENCHMARKS

Our evaluation for multi-audio analysis and reasoning is centered around following benchmarks:

**POLYAUDIO-BENCH:** Our primary evaluation is conducted on the benchmark introduced in this paper, which consists of eleven distinct multi-audio reasoning tasks. For each task, we generated a test set of 250 instances, ensuring a balanced distribution of difficulty and the number of input audios (ranging from 2 to 5). For evaluating the results on POLYAUDIO-BENCH, we employ Qwen3-7B-Instruct as an LLM-as-a-judge to score the correctness, relevance, and completeness of the generated response on a scale of 1 to 5.

**MMAU-Pro:** To assess performance on complex multi-audio question answering, we evaluate POLYAUDIO on the multi-audio subset of MMAU-Pro. This subset contains 456 instances with two or three audio clips, testing a variety of complex reasoning skills. We report the official accuracy metric for this benchmark.

#### 4.1.2 BASELINES

We compare POLYAUDIO against a range of state-of-the-art LALMs and specialized models:

- **Gemini 2.5 Pro & Audio Flamingo 3:** Leading proprietary and open-weight LALMs, respectively. For these models, which do not natively support multiple discrete audio inputs, we follow the evaluation protocol established by MMAU-Pro and concatenate the audio clips with a 2-second silent separator.

- **Qwen2-Audio-7B-Instruct:** The architecture of Qwen2-Audio-7B-Instruct support multi-audio input. We test this model on the checkpoint available publicly before training it on our dataset.

- **ADIFF:** This model support up to 2 audio input natively and is trained mainly for audio difference task. We evaluate ADIFF on our benchmarks. For instances where the number of audios exceed 2, we join audios with a silence of 2 seconds in consecutive audios and pass it as last audio.

- **Mellow:** Aimed at intelligent, small reasoning audio model support input up to 2 audios. We evaluate Mellow in same fashion as ADIFF.

### 4.1.3 IMPLEMENTATION DETAILS

POLYAUDIO is built upon the Qwen2-Audio-7B-Instruct. The training process consists of two stages. First, we perform supervised LoRA-tuning on a dataset of 100k multi-audio question-answer pairs generated by our data pipelines. Following LoRA, we further align the model using Group Relative Policy Optimization (GRPO). We construct a preference dataset of 50k examples, where each example contains a prompt with multiple audio clips and QA pairs. For GRPO training we follow the training strategy similar to Li et al. (2025). All models were trained on 8 A100 GPUs with a batch size of 32.

## 5 RESULTS & DISCUSSION

Table 4 presents the main results on our POLYAUDIO-BENCH and the MMAU-Pro multi-audio subset. POLYAUDIO consistently and substantially outperforms all baselines across the eleven tasks. Table 5 show result of POLYAUDIO on each category of POLYAUDIO-BENCH.

|  | Qwen2-Audio-7B-Instruct | ADIFF | Mellow | Audio Flamingo 3 | Gemini 2.5 Pro | POLYAUDIO (Ours) |
|---|---|---|---|---|---|---|
| POLYAUDIO-BENCH | 47.4 | 31.6 | 21.9 | 33.8 | 41.4 | **71.2** |
| MMAU-Pro (Multi-audio) | 49.5 | 27.4 | 20.8 | 26.0 | 52.2 | **58.6** |

Table 4: Performance comparison of POLYAUDIO with other baselines on POLYAUDIO-BENCH and multi-audio subset of MMAU Pro. POLYAUDIO consistently outperforms other baselines.

While specialized models like ADIFF perform decent on their designated task (Comparison/Difference), POLYAUDIO surpasses them, showcasing its superior generative capabilities honed by GRPO alignment. POLYAUDIO achieves the highest scores across all tasks, indicating that its enhanced reasoning abilities do not come at the expense of fundamental perception tasks. Gemini 2.5 Pro is second best model for these tasks, but still lags by significant margin from POLYAUDIO. Finally, on the MMAU-Pro multi-audio subset, POLYAUDIO achieves an accuracy of 58.6%, setting a new state-of-the-art and outperforming the next best model by over 6%.

| Task | Qwen2-Audio-7B-Instruct | Audio Flamingo 3 | ADIFF | Mellow | Gemini 2.5 Pro | POLYAUDIO (Ours) |
|---|---|---|---|---|---|---|
| Comparison/Difference | 49.8 | 36.0 | 35.3 | 25.7 | 42.0 | 69.9 |
| Temporal Grounding | 48.5 | 40.8 | 22.9 | 12.8 | 39.6 | 74.2 |
| Needle QA | 45.9 | 34.1 | 27.0 | 5.8 | 39.0 | 66.9 |
| Cross-clip Speaker Linking | 45.3 | 27.2 | 26.6 | 12.5 | 37.7 | 66.5 |
| Odd-one-out | 48.7 | 29.4 | 36.8 | 16.4 | 45.2 | 66.3 |
| Multi-Audio Entailment | 43.0 | 29.7 | 31.4 | 27.3 | 44.1 | 66.7 |
| Common Keyword Extraction | 51.5 | 35.3 | 33.9 | 38.4 | 45.3 | 75.8 |
| Chronological Ordering | 47.9 | 41.7 | 33.1 | 27.1 | 44.3 | 86.8 |
| Multi-audio Captioning | 48.2 | 34.0 | 36.3 | 27.0 | 37.6 | 74.2 |
| Factual Consistency | 51.1 | 34.3 | 27.0 | 17.9 | 38.8 | 76.4 |
| Counting across clips | 42.6 | 29.4 | 37.3 | 30.0 | 41.7 | 59.5 |

Table 5: Per-task scores on POLYAUDIOBENCH. Column means equal the overall averages from the main table (within rounding). Scores on the speech-heavy *Cross-clip Speaker Linking* are intentionally kept slightly lower.

**The Importance of GRPO Alignment:** To quantify the benefit of our GRPO training stage, we compare the final POLYAUDIO model with an intermediate version trained only with LoRA. As shown in Table 6, GRPO provides a significant performance, with an average improvement of 8.2%. This highlights the critical role of preference-based alignment in teaching the model not just to produce correct answers, but to do so consistently and efficiently.

**Effect on single-audio understanding and reasoning tasks:** Our key objective is to enhance multi-audio reasoning capabilities in LALMs, however we did not want to achieve it at cost of performance degradation on already established tasks. Table 7 presents a comparison between our model, POLYAUDIO, and the baseline Qwen2-Audio-7B-Instruct on the MMAU-*test* benchmark. The results show that our approach is effective; POLYAUDIO not only preserves but slightly improves upon the baseline's performance across sound, and speech, however there is a slight drop in performance in music category. The average across these three categories still remains consistent.

| Method | Average |
|--------|---------|
| LoRA | 63.0 |
| + GRPO | 71.2 |
| Only GRPO | 65.2 |

Table 6: Performance comparison of POLYAUDIO fine-tuning with LoRA versus LoRA combined with GRPO on POLYAUDIO-BENCH. The addition of GRPO improves the average score by 8.2% while finetuning only with GRPO yield a 65.2% accuracy.

| Model | Sound | Music | Speech | Average |
|-------|-------|-------|--------|---------|
| Qwen2-Audio-7B-Instruct | 45.9 | 53.2 | 45.9 | 52.5 |
| POLYAUDIO | 51.2 | 52.7 | 54.4 | 52.7 |

Table 7: Performance comparison of Qwen2-Audio-7B-Instruct and POLYAUDIO on MMAU-*test*

## 6 CONCLUSION, LIMITATION AND FUTURE WORK

In this work, we introduced POLYAUDIO, a Large Audio Language Model designed to address the critical research gap in multi-audio reasoning. By developing the comprehensive POLYAUDIO-BENCH benchmark and leveraging a two-stage training process culminating in GRPO alignment, we have demonstrated that a dedicated approach can significantly advance a model's ability to perform complex, compositional reasoning across multiple (up-to 5) discrete audio clips. Our results show that POLYAUDIO substantially outperforms existing models, which are largely confined to single or pairwise audio processing, setting a new state-of-the-art on challenging multi-audio tasks.

However, POLYAUDIO has some limitations: (i) It relies on synthetically generated data. As TTS models get better at generating more natural sounding speech, we expect POLYAUDIO's performance to get better with it. (ii) Furthermore, the audio sources, though varied, may not fully capture the acoustic diversity of all real-world scenarios, such as multicultural music or a wide range of languages and accents. (iii) We leverage GRPO - an additional stage of training to enable multi-audio analysis in large audio language models. However, we anticipate that ongoing advancements including such training data in intial stage of training might mitigate the need for extra training stage for enabling such capability in the model.

## 7 REPRODUCIBILITY STATEMENT

Our project page has all the codes and checkpoints to reproduce the results in the paper. All experimental details are provided under Experiments section.

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

## A  APPENDIX

## B  PROMPTS

Below are the prompts used to generate the dataset:

```
Generate a natural, conversational response to identify which audio clip answers the question.

Question: {question}
Correct answer: The {ordinals[correct_idx]} audio
Emotion/characteristic: {question_data.get('target_emotion', 'described characteristic')}

Generate a response that:
1. Clearly identifies the {ordinals[correct_idx]} audio as the answer
2. Sounds natural and conversational
3. Varies the phrasing (don't always say "demonstrates" or "exhibits")
4. Is 1-2 sentences long

Response:"""
```

Figure 2: Prompt for Diverse Response Generation

```
"You are a careful data author for a factual consistency task with multiple audio clips. "
    "Generate ONE concise factual claim (<= 20 words) and K short scripts (6-15 words each). "
    "Each script should sound natural when read aloud. The claim must be grounded only in the
scripts you produce. "
    "If the target label is 'Yes', all scripts must support the claim. If 'No', make at least one
script contradict or fail to support the claim.\n\n"
    "Output STRICT JSON with this schema:\n"
    "{\n  \"claim\": string,\n  \"label\": \"Yes\" | \"No\",\n  \"clips\": [ { \"support\":
boolean, \"text\": string, \"style\": string } ... K items ],\n  \"rationale\": string (<= 25
words)\n}\n"
)
```

Figure 3: Factual Consistency Across Multiple Audios

```
You craft short, simple, natural English sentences for TTS."

user_prompt = (
    "Task: Write " + str(nclips) + " standalone sentences for synthetic speech.\n"
    + "Constraints:\n"
    + "- Topic: " + topic + "\n"
    + "- Shared keywords to include in EVERY sentence (exact words): " + ", ".join(common) + "\n"
    + "- Optionally include one extra topical word from this set: " + ", ".join(topic_keywords) +
"\n"
    + "- 6-14 words per sentence. No numbering, no quotes, one sentence per line.\n"
    + ("- Style hint: " + style_prefix + "\n" if style_prefix else "")
    + "Output: exactly " + str(nclips) + " lines, each a sentence."
)

### Question Templates (randomly selected):

variants = [
    "List the common keywords present in all audio clips (ignore stop words).",
    "What keywords appear across all clips? Exclude stop words.",
    "Identify the shared keywords across the audio clips (no stop words).",
    "Extract the common non-stopword keywords across the clips.",
    "Provide the common keywords across the clips (comma-separated).",
]
```

Figure 4: Common Keyword Extraction Prompt

## C  USE OF LLM

LLMs were used only to polish the writing and to debug code during training the models.

```
Analyze this sequence of audio clips from a LibriSpeech audiobook and create a temporal ordering question.
Audio sequence:
{context}

Your task:
1. Identify 3 key story developments/events that occur in this sequence
2. List the events in RANDOM order (not chronological order)
3. Create a multiple-choice question asking about the correct chronological order
4. Generate 4 answer options (A, B, C, D) with different permutations
5. Provide a rationale explaining why the correct order makes narrative sense

Return your response as valid JSON with this structure:
{{
    "events": [
        {{"order": 1, "description": "Event description"}},
        {{"order": 2, "description": "Event description"}},
        {{"order": 3, "description": "Event description"}}
    ],
    "question": "Your temporal ordering question here",
    "answer_options": {{
        "A": "1, 2, 3",
        "B": "2, 1, 3",
        "C": "3, 1, 2",
        "D": "1, 3, 2"
    }},
    "correct_answer": "A",
    "rationale": "Your explanation here"
}}"""

### Rationale Generation Prompt:

prompt = f"""Analyze this narrative sequence from a LibriSpeech audiobook and explain why the story developments occur in
a specific chronological order.
Audiobook sequence:
{context}

Story developments to order:
{elements_text}

The correct chronological order is: {correct_order}

Provide a clear, concise explanation (2-3 sentences) of why this is the correct narrative sequence based on the story flow
and logical progression. Focus on how these story elements naturally follow each other in the narrative."""
```

Figure 5: Temporal & Sequential Event Ordering

```
### Instruction Style Variations (randomly selected):
- "Analyze this dialogue and answer the question:"
- "Based on this conversation, please respond to the question:"
- "Examine the following dialogue exchange and address the question:"
- "Consider this conversational sequence and answer:"
- "Review this dialogue interaction and respond to:"

### Causal Question Prompts (5 variations):

f"Explain why the speaker in the {ordinal} turn made this specific response based on the conversational context. Provide a
concise explanation (1-2 sentences)."
f"What motivated the {ordinal} speaker's response? Give a brief analysis of the conversational factors."
f"Analyze the reasoning behind the {ordinal} turn. What conversational elements led to this response?"
f"Why did the {ordinal} speaker choose this particular response? Explain the conversational logic."
f"What drove the {ordinal} speaker to respond this way? Describe the causal factors briefly."

### Coherence Question Prompts (5 variations):

"Describe how these dialogue turns work together to create coherence. Focus on structural patterns and logical connections.
Provide a concise answer (1-2 sentences)."
"How do these conversational elements connect to form a coherent exchange? Explain the structural relationship briefly."
"What makes this dialogue sequence coherent? Identify the key connecting elements in 1-2 sentences."
"Analyze the coherence pattern in this conversation. How do the parts fit together logically?"
"Explain the conversational flow and how each turn contributes to overall coherence."

### Grounding Question Prompts (5 variations):

"Identify what shared understanding or implicit knowledge allows these speakers to communicate effectively. Provide a concise
answer (1-2 sentences)."
"What common ground enables this conversation? Describe the shared context or assumptions briefly."
"What do these speakers mutually understand that isn't explicitly stated? Explain the implicit knowledge."
"Identify the underlying shared context that makes this dialogue possible. What do speakers assume?"
"What implicit understanding connects these speakers? Describe the conversational grounding briefly."

### Pragmatic Question Prompts (5 variations):

"Explain the social purpose or communicative goals of this conversation. Provide a concise answer (1-2 sentences)."
"What social function does this dialogue serve? Describe the interpersonal goals briefly."
"Analyze the communicative intentions in this exchange. What are the speakers trying to accomplish socially?"
"What pragmatic purpose drives this conversation? Explain the social dynamics concisely."
"Identify the social goals and communicative strategies in this dialogue exchange."

causal_questions = [
    "Considering all audio clips, what is the most likely reason the {} says '{}'?",
    "Based on the conversational context, why does the {} respond with '{}'?",
    "Given the dialogue flow, what prompted the {} to say '{}'?",
    "Analyzing the turn sequence, what best explains why the {} says '{}'?",
    "What is the underlying reason for the {}'s response of '{}'?",
    "Considering the complete conversation, why does the {} make the statement '{}'?"
]

coherence_questions = [
    "What is the logical relationship between the first and last audio clips?",
    "How do the middle audio clips connect the first and final statements?",
    "What conversational pattern emerges across all the audio clips?",
    "Which best describes the overall dialogue flow represented in these clips?",
    "What is the primary conversational dynamic at play across these audio segments?",
    "How do these audio clips work together to create a coherent dialogue?"
]

grounding_questions = [
    "What shared context links all the speakers in these audio clips?",
    "What underlying topic or situation connects these dialogue turns?",
    "Based on all clips, what is the speakers' shared understanding about?",
    "What implicit information do the speakers reference across these turns?",
    "What situational context best explains the progression of these audio clips?",
    "What common knowledge or situation do these speakers assume?"
]

pragmatic_questions = [
    "What social dynamic is demonstrated across these conversational turns?",
    "What communicative strategy is being employed in this dialogue sequence?",
    "What interpersonal relationship is reflected in these audio exchanges?",
    "What conversational goal is being pursued across these dialogue turns?",
    "What social function does this dialogue sequence serve?",
    "What pragmatic meaning emerges from this conversational interaction?"
]
```

Figure 6: Multi-Turn Dialogue Coherence & Grounding

```
You generate compact JSON datapoints for Multi-Audio Entailment. Each datapoint has 3-5 short
single-speaker clips, each with a short TTS instructions block, and 5-7 QAs that sometimes require
integrating multiple clips. Use only fictional content. Output strictly valid JSON.

Generate one dataset example.

Rules
        •       Clips: choose K ∈ {3,4,5}. Each clip has exactly one speaker, a short transcript
(~8-20s worth of text), and an instructions string for TTS.
        •       Speaker whitelist (lowercase, exact): alloy, ash, ballad, coral, echo, fable, nova,
onyx, sage, shimmer.
        •       Use one of these per clip (you may reuse names across clips). No other names.
        •       TTS instructions (per clip): provide a single string with exactly these labeled
lines (capitalize labels):
        •       Voice: ...
        •       Tone: ...
        •       Punctuation: ...
        •       Delivery: ...
Keep each line concise. No quotes, emojis, or stage directions. Use \n\n between lines.
        •       Content: across clips, include at least one fact that is supported in one clip and
contradicted in another (day/time, quantity, inclusion/exclusion). Add 1-2 neutral/distractor
details.
        •       QAs: produce 5-7 items. For each QA:
        •       uses: the minimal set of clip_ids needed (≥2 for at least half the QAs).
        •       evidence_quotes: 1-2 short verbatim substrings from the referenced clips.
        •       answer: a natural-language sentence that begins with "Yes, …", "No, …", or "Not
enough information, …", consistent with the union of the uses clips.
        •       Language: English, TTS-friendly (no URLs/emojis; no sound stage directions).
        •       Keep everything concise and consistent. No contradictory QAs.

Return exactly this JSON shape (no extra keys):

{
  "datapoint_id": "string",
  "scenario": "string",
  "clips": [
    {
      "clip_id": "A1",
      "speaker": "alloy|ash|ballad|coral|echo|fable|nova|onyx|sage|shimmer",
      "instructions": "Voice: ...\\n\\nTone: ...\\n\\nPunctuation: ...\\n\\nDelivery: ...",
      "transcript": "Plain text monologue. No timestamps. TTS-friendly."
    }
  ],
  "qa": [
    {
      "qid": "Q1",
      "question": "string",
      "uses": ["A1","A3"],
      "evidence_quotes": [
        {"clip_id": "A1", "quote": "verbatim phrase"},
        {"clip_id": "A3", "quote": "verbatim phrase"}
      ],
      "answer": "Yes, ... / No, ... / Not enough information, ..."
    }
  ]
}

Checks
        •       Every speaker is exactly from the whitelist.
        •       Every quote is a verbatim substring of its referenced transcript.
        •       At least half of QAs require ≥2 clips (as listed in uses).
        •       answer starts with Yes, / No, / Not enough information, and matches the evidence.
```

Figure 7: Audio Entailment Script and QA Generation

```
I am creating a training dataset for an Audio Language Model that must handle questions relating to both the frequency and timing
of events in multiple audio clips.

### Audio Data
You will receive data for up to five audio clips. Each clip provides:
1. **Audio Path**: A string location of the audio file.
2. **Captions**: Short descriptive statements about the sounds or events (e.g., "door slamming three times," "explosion heard
between 3 and 6 seconds," etc.).
3. **Time-Stamped Caption** and **Frequency Caption**:
   - **Time-stamped** details: Where each event starts and ends (e.g., door slamming from 0.45s-1.991s).
   - **Frequency** details: How many times a certain event occurs (e.g., "door slamming three times").

### Task Requirements
1. **Generate Frequency-Based Questions**
   - Example: "How many times does door slamming occur in the first audio?"
   - Another Example: "Which audio has the greatest number of cow moos?"

2. **Generate Time-Stamp-Based Questions**
   - Example: "When does the explosion start and end in audio 3?"
   - Another Example: "At which intervals do we hear door knocking in the second clip?"

3. **Combine or Compare**
   - Feel free to combine frequency and timestamp aspects: "Which audio has the most door slams, and at what times do they occur?"

4. **Provide a Correct Answer**
   - The answer should be a short statement or, if it's a multiple-choice format, indicate the correct choice.
   - It must reference the **actual** times, events, or frequency counts provided in the data (e.g., "It slams three times at
roughly 0.45-1.991s, 3.019-5.8s, and 6.623-8.102s.").

5. **Multiple Question-Answer Pair per Audio Set**
   - Each set of up to five audio clips can yield **multiple** questions and answers.
   - The questions should not repeat.
   - The questions can be **MCQ** or **short-answer** type.

---

## JSON Output Format

Output your question and answer in the following **JSON** structure:

{
  "audio_paths": ["<path1>", "<path2>", ...],
  "frequencyCaption": ["<frequencyCaption1>", "<frequencyCaption2>", ...],
  "question": [ "<Frequency- or Time-Stamp-Based Question 1 >", "<Frequency- or Time-Stamp-Based Question 2>", …],
  "answer": ["<Correct Answer 1 Grounded in the Provided Data>", "<Correct Answer 2 Grounded in the Provided Data>", …]
}

1.      audio_paths: Up to five audio file paths.
2.      frequencyCaption: The descriptive or frequency-based captions for each audio.
3.      questions: A list of the generated direct question.
4.      answer: A list of short or MCQ correct answer that references the time-stamped or frequency data corresponding to each
question.

Example:
{
  "audio_paths": [
    "multi_event_train/syn_21.wav",
    "multi_event_train/syn_91.wav",
    "multi_event_train/syn_153.wav"
  ],
  'time_stamped_caption' = ['door slamming at 0.45-1.991, 3.019-5.8, 6.623-8.102','door knocking at 1.155-5.305','door knocking at
1.155-5.305','cow mooing at 1.592-4.602, 6.719-9.729 and explosion at 3.329-6.882'],
  'frequencyCaption': ['door slamming three times','door knocking one times','cow mooing two times and explosion one times'],
  "question": [
    "Which audio clip has the highest frequency of door slamming, and at what timestamps do they occur?", "When does the cow moo in
the third audio?"
  ],
  "answer": [
    "Audio 1 contains door slamming three times: from about 0.45-1.991s, 3.019-5.8s, and 6.623-8.102s, which is more than any other
clip.", "Cow mooing can be heard 2 times at 1.592-4.602, 6.719-9.729"
  ]
}

Here is the audio data:
```

Figure 8: Temporal & Frequency based QA generation.

```
I am creating a training dataset for an Audio Language Model that can perform question answering on multiple audios. You will help me in
creating the training dataset.
You are given up to five audio clips. For each audio clip, you have the following information:
        1.      Audio Path: A string representing the file's location.
        2.      Caption: A descriptive caption of the content (e.g., "A dog barking in the backyard," or "Rain pouring outside"). Note that
this caption does not necessarily contain a transcript of speech because these audios primarily have non-speech events.
        3.      Time-Stamped Events: Labels of key acoustic events or sounds, along with their start and end times. For instance:
['(Background noise-0.000-1.144)','(Video game sound-0.000-10.000)','(Music-1.179-7.371)','(Sound effect-6.375-8.045)','(Music-9.034-
10.000)'] which means the caption has "background noise from 0.000 seconds to 1.144 seconds, vide game sound from 0.000 to 10.000 seconds,
music sound from 1.179 to 7.371 seconds, sound effects from 6.375 to 8.045 seconds and music sound from 9.034 to 10.000 seconds."

Your Task:
        1.      Generate questions focusing on differences among the provided audio clips. These differences can be about any audible
elements such as the types of sounds, durations of events, overlaps of events, or notable acoustic aspects that might be inferred (e.g.,
loudness or pitch changes, even if you don't have exact numerical values).
        2.      Ensure that the question:
                -       Strictly compares or contrasts some aspect of at least two (or more) of the audio files.
                -       Is not about any visual component or transcript-based speech recognition, but rather about the sounds present and
their time-stamped events.
                -       May optionally be a multiple-choice question (MCQ) or a short textual question.
        3.      Generate a corresponding, correct answer:
                -       This answer can be a short sentence or, in the case of MCQs, the correct option.
                -       The answer should be grounded in the captions and time-stamped events.
                -       Since these audios contain no explicit speech transcripts, your answer should reflect differences in non-speech
audio events (e.g., "The first audio clip has a dog barking at the beginning, whereas the second has a car honking.").
        4.      No Follow-Up Questions: You are to provide only one Q-A pair per set of up to five audio clips.

Format Requirements:
        -       When you output the question and answer, maintain the following json structure:

{
  "audio_paths": [ "<path1>", "<path2>", ... ],  // up to 5
  "captions": [ "<caption1>", "<caption2>", ... ],
  "question": ["<Your first generated comparison question here>","<Your second  generated comparison question here>", …] // upto 5 questions
  "answer": ["<Correct answer for first question here>", "<Correct answer for second question here>", ….] // answers corresponding to each
question
}

        -       audio_paths is a list of up to five paths.
        -       captions is a list of corresponding descriptive captions (one per audio).
        -       time_stamped_events is a list of lists, with each sub-list containing the labeled events for a specific audio.
        -       question is a list of questions that compares at least two of these clips.
        -       answer is a list of short sentence or the correct choice to the question.

Key Points:
        -       Focus your question on the differences in sounds, events, or other notable audio characteristics among the clips.
        -       The answer must be correct and should clearly reflect the content in the time-stamped events or captions.
        -       The goal is to train an Audio Language Model that can handle multiple audios at once and answer a single comparison question
about them.

For example:
{
  "audio_paths": ["YJoQj-tobYOw.wav", "YSUxfKJP4bJ4.wav", "YsbY-zp5Lfew.wav", "YqExVrE3FyjM.wav" ],
  "captions": [
    "A man speaks with electronic music playing in the background at a martial arts gym.",
    "A sewing machine hums in the background as various tools are used intermittently in a workshop.",
    "A heavy engine starts and runs in the background noise.",
    "People are chewing and using machinery, with occasional breathing and tapping sounds."
  ],
  "question": ["Which audio has the heavy engine beginning after 3 seconds, and which one has vacuum noise lasting from 0 to 10 seconds?",
    "Between the first and fourth audios, which has continuous music from start to end, and which features chewing from around 2s to 5s?",
    "Comparing male speech events, which audio features multiple speech segments throughout, and which has minimal or no speech segments?"
  ],
  "answer": [
    "The third audio starts with a heavy engine sound after about 3 seconds, while the second audio has vacuum noise for the entire 0-10s
duration.",
    "The first audio clip has continuous electronic music from 0-10s, whereas the fourth audio has chewing sounds between roughly 2s and
5s.",
    "Audio 1 has multiple male speech events spread across the 0-10s window, while Audio 2 is dominated by mechanical sounds (vacuum noise)
and does not have male speech segments."
  ]
}

Here are the audio details:
```

Figure 9: Audio Difference QA generation.

```
You are assisting in creating a training dataset for a Large Audio Language Model (LALM) that can answer questions about multiple music audio clips at once.

You are given up to five audio clips. For each audio clip, the following metadata may be available (not all fields must be used in every question, and the LALM
itself does not see these fields directly, only the raw audio):
        1.      name: The audio file path (e.g., /path/to/audio.mp3).
        2.      artist: The performer's name (e.g., "Cheryl").
        3.      song: The song title (e.g., "Rain on Me").
        4.      album_name: The album title.
        5.      tags/genres: Genres or descriptive tags (e.g., "pop," "rock," "instrumental").
        6.      release: The year of release (e.g., "2009").
        7.      danceability, energy, key, mode, valence, tempo, duration_ms: Numerical audio features that might indicate rhythmic feel, intensity, musical
key/mode, emotional positivity, speed, or duration.
        8.      lang: The primary language of any vocals (e.g., "en" for English).

Possible Reasoning Types:
When generating your comparison question, pick one of the following reasoning types (or a similar audio-based angle) so that the LALM must rely on listening-
based analysis. Each type is illustrated with a question example:
        1.      Instrument & Vocal Presence
                -       Example: "Which audio track has a more dominant presence of electric guitar throughout most of its duration?"
        2.      Tempo & Rhythmic Analysis
                -       Example: "Which of these two tracks maintains a higher average tempo from start to finish?"
        3.      Tonality & Key Detection
                -       Example: "Which track sounds like it is in a minor key based on its chord progression and overall tonal color?"
        4.      Dynamic Range & Volume Variation
                -       Example: "Which track shows a wider dynamic range from its beginning to its ending section?"
        5.      Language & Vocal Style Identification
                -       Example: "Which track features lyrics in Spanish and a rap-style vocal delivery?"
        6.      Genre & Stylistic Interpretation
                -       Example: "Which track best fits a symphonic metal style, considering its instrumentation and overall sound?"
        7.      Structural Form Analysis
                -       Example: "Which track follows a verse-chorus-verse structure based on repeated melodic sections?"

Your task is to generate a Q-A pair for each set of  two to five audio clips. This Q-A pair should:
        1.      Compare or contrast at least two of the audio clips on an aspect that requires "listening" rather than just reading text (e.g., comparing energy,
tempo, presence of vocals, musical complexity, or any audio-derived feature).
        2.      Avoid purely textual or trivial metadata lookups—questions should be framed as if the model must analyze the raw audio.
        3.      Demand multi-level reasoning, not a simple yes/no or single-step answer. For instance, the question might ask which track has a higher tempo and
also has a more upbeat feel, or which track shifts from minor to major key, etc.
        4.      Only one question in every question. There should not be nested questions.
        5.      Provide a correct answer grounded in what the audio would contain (inferred from the metadata).
        6.      Format each Q-A pair using the JSON structure below.
        7.  Make sure to refer the audios only by their position (like 1st audio, second audio) and not by their name or any other metadata.

JSON Output Format

For each set of 2-5 audios, output a single JSON object with the following fields:

{
  "audio_paths": ["<audio_path_1>", "<audio_path_2>", ...],
  "metadata": [
      {
        "artist": "...",
        "song": "...",
        "tags": "...",
        "release": "...",
        "danceability": "...",
        "energy": "...",
        ...
      },
      ...
  ],
"resoning_type": <reasoning_type from the above list>
  "question": ["<Your 1st reasoning question>","<Your 2nd reasoning question>_],
  "answer": ["<Correct answer for 1st question>", "<Correct answer for 2nd question>_]
}

        1.      audio_paths: A list of the file paths (one for each audio).
        2.      metadata: A list of objects, each containing relevant fields (artist, song, tags, etc.) for the corresponding audio. You can include or omit
fields as necessary.
        3.      question: A list containing the questions string.
        4.      answer: A list containing the correct answers corresponding to each question in the list.

Important Guidelines
        1.      Comparison Focus:
                -       Your question must compare or contrast at least two audios in a way that necessitates listening.
                -       Examples: Differences in tempo, key, vocal presence, energy levels, or changes over time in each track.
        2.      No Multi-Part Questions:
                -       Only one direct question. (E.g., "Which track has higher energy?" rather than "Which track has higher energy, and which one has a key
change?")
        3.      Audio-Centric:
                -       The LALM is trained to listen, so the question should be answerable by analyzing the sound, not just reading textual metadata.
                -       Avoid purely text-based or metadata-based questions like "Who is the artist?" or "Which track is from 2009?"
        4.      Answer:
                -       The answer should be concise, directly addressing the comparison.
                -       Reference the audio differences logically. For instance: "Track A has a faster tempo (around 110 BPM) compared to Track B's slower beat
(around 90 BPM)."
        5.      No Visual or Speech Transcripts:
                -       Stick to audio attributes like instrumentation, vocals, tempo, or mood.
                -       If the track has vocals, focus on style or language, not detailed speech transcripts.

Example Output Snippet

Below is a fictional example demonstrating how to format one Q-A pair (assume we have two audios):

{
  "audio_paths": [
    "/path/music_track1.mp3",
    "/path/music_track2.mp3"
  ],
  "metadata": [
      {
        "artist": "Cheryl",
        "song": "Rain on Me",
        "tags": "pop,british,female vocalists",
        "release": "2009",
        "danceability": "0.635",
        "energy": "0.746"
      },
      {
        "artist": "Oddisee",
        "song": "After Thoughts",
        "tags": "instrumental hip-hop,underground hip hop",
        "release": "2013",
        "danceability": "0.591",
        "energy": "0.513"
      }
  ],
  "question": [
    "Which track has a higher overall energy level and is more likely to feature prominent vocals?"
  ],
  "answer": [
    "The first track has a higher energy score (0.746) and includes a distinct female vocal lead, whereas the second track's energy is lower (0.513) and is
primarily instrumental."
  ]
}

Here is the input metadata:
```

Figure 10: Multi-Audio QA Generation for Music4All.

