# OpenReview forum: "PolyAudio: Advancing Multi-Audio Analysis & Reasoning in Large Audio Language Models"
_ICLR.cc/2026/Conference — ICLR 2026 Conference Withdrawn Submission_

### Official Review · Reviewer_T4tF · 2025-10-17

**Soundness:** 2
**Presentation:** 2
**Contribution:** 2
**Rating:** 4
**Confidence:** 4

**Summary:**

This paper presents POLYAUDIO, a large audio-language model (LALM) designed for reasoning across multiple audio clips (up to five). The authors introduce a new benchmark, POLYAUDIO-BENCH, covering 11 distinct reasoning categories, and propose a two-stage training framework that combines LoRA fine-tuning with Group Relative Policy Optimization (GRPO) alignment. Experimental results show that POLYAUDIO significantly outperforms Qwen2-Audio, Audio Flamingo 3, and Gemini 2.5 Pro on multi-audio reasoning benchmarks.

**Strengths:**

(1) The paper targets an underexplored yet important problem of multi-audio compositional reasoning, extending beyond the conventional pairwise paradigm in audio-language models.

(2) The proposed benchmark POLYAUDIO-BENCH is comprehensive and systematically designed reasoning tasks.

**Weaknesses:**

(1) While POLYAUDIO itself natively supports discrete multi-clip inputs, baselines use concatenation for evaluation, which complicates fair comparison on compositional reasoning.

(2) The paper claims to interleave audio embeddings with text tokens, but never describes how positional encoding, cross-clip ordering, or temporal alignment are handled. Without explicit fusion mechanisms (e.g., cross-attention across clip embeddings or learned temporal gates), the model likely treats multi-clip inputs as a bag-of-embeddings, which weakens compositional reasoning fidelity.

(3) Unlike recent Audio LLMs (e.g., Qwen2.5-Omni, Audio Flamingo 3) that introduce contrastive or embedding-consistency losses to stabilize audio–text fusion, POLYAUDIO relies purely on LoRA + GRPO without auxiliary alignment. This design risks latent drift between audio and language representations, especially under synthetic prosody conditions.

(4) Evaluation covers only synthetic and benchmark data, without tests on human-recorded or cross-domain audio.

(5) The paper omits any mention of temporal windowing, stride length, or max tokenization limit. This makes it unclear whether temporal grounding tasks rely on genuine reasoning or implicit positional priors learned from synthetic prompts.

(6) Using Qwen3-7B-Instruct as a scalar scorer measures lexical/semantic similarity rather than true causal or compositional reasoning accuracy. This conflates linguistic fluency with reasoning validity, which is particularly problematic for audio-based tasks like counting or factual consistency.

(7) The paper lacks examples of reasoning errors, such as miscounting, misattributed speakers, or cross-clip hallucination.

(8) The preference dataset is auto-generated by GPT-5, but there’s no evidence of label noise filtering or inter-sample consistency checks. No statistics on reward variance, prompt diversity, or reward drift are reported.

Typo:

Line 50: invalid citation

**Questions:**

(1) Since POLYAUDIO supports discrete multi-clip inputs but several baselines (e.g., ADIFF, Mellow) are evaluated using concatenated audio with 2-second silence, could the authors clarify how this affects fairness and comparability? Have you considered an ablation where all models are tested under the same concatenation protocol to quantify the potential advantage?

(2) The paper states that audio embeddings are interleaved with text tokens, but it remains unclear how positional encoding, clip ordering, or temporal alignment are handled internally. Could the authors elaborate on whether any cross-clip attention or learned gating mechanism is used to preserve temporal structure and compositional dependencies across clips?

(3) POLYAUDIO relies solely on LoRA fine-tuning followed by GRPO alignment, whereas recent Audio LLMs employ additional contrastive or consistency regularization. Have the authors tested whether adding a lightweight cross-modal loss (e.g., audio–text contrastive alignment) improves stability or mitigates latent drift?

(4) Given that both training and evaluation rely on GPT- and TTS-generated data, can the authors assess POLYAUDIO on any human-recorded or real-world acoustic benchmarks (e.g., VoxConverse, AVA-Speech)? Such results would strengthen claims about real-world reasoning generalization.

(5) The model processes up to five clips of 30 s each, but implementation details such as window size, stride, and tokenization limit are not described. Could the authors clarify how long clips are segmented and whether the model encodes explicit temporal order or duration information during reasoning?

(6) Since Qwen3-7B-Instruct serves as the sole judge, how do the authors ensure that the metric captures reasoning correctness rather than linguistic fluency or paraphrase similarity? Would you consider adding human evaluations or multiple LLM-as-a-Judge setups to assess correlation and robustness?

(7) The paper reports only aggregate scores without showing examples of reasoning errors or failure cases. Could the authors provide qualitative analyses (e.g., miscounting, cross-clip confusion, hallucinated entities) to illustrate current limitations and guide future improvements?

(8) For the GPT-5-generated POLYAUDIO-PREF dataset, have the authors measured label noise, prompt diversity, or reward variance? Including such diagnostics or filtering criteria would clarify the stability of GRPO training and the reliability of the reward signal.

---

### Official Review · Reviewer_ov1n · 2025-10-26

**Soundness:** 1
**Presentation:** 2
**Contribution:** 1
**Rating:** 2
**Confidence:** 3

**Summary:**

This paper studies the problem of multi-audio understanding and reasoning (i.e., between 2 and 5 clips). They propose Polyaudio, a LALM specifically tailored for this task which is trained via SFT and GRPO. They also provide two curated datasets, PolyAudio-Train and PolyAudio-Pref, to train their proposed method, and a benchmark to test multi-clip reasoning. Evaluations on PolyAudio-Bench and a subset of MMAU-Pro reveals the superiority of Polyaudio over the other baselines.

**Strengths:**

- The paper is clearly written.
- The objectives of the papers and the main contributions are clearly described.
- The problem of multi-audio reasoning is of sufficient significance.

**Weaknesses:**

- The authors claims at lines 88-90: “We introduce POLYAUDIO, a novel Large Audio Language Model specifically designed for complex reasoning across multiple (up-to 5)…”. However, the proposed model does not bring any novelty. In fact, it simply concatenates the audio tokens from different clips and feed them to the LLM. It is a straightforward idea and I don’t see any novelty on doing this, just a natural way of processing multiple clips. In addition to this, it simply uses a SFT stage followed by GRPO alignment, a common strategy in the literature.
- The Polyaudio-train and -pref datasets are a collection of existing datasets + syntethic speech generated using LLMs. It is not clear how much data comes from each dataset and from the synthetic ones. It is not clear to me why the authors have decided to generate synthetic data, they should elaborate more on this.
- The authors do not include any real examples coming from the proposed training and evaluation benchmarks to understand better the tasks the models need to reason about.
- I don’t think it’s fair to compare a model which follows SFT + GRPO alignment on 150k data on the task of multi-clip reasonings with models which have been trained on single clips.
- The model is only tested on multi-audio tasks, however it should be also tested on single clips reasoning. Does the model still retain good performance on these tasks or is it subject to catastrophic forgetting?
- I would expect the authors to at least mention that the datasets Polyaudio-Train and -Pref will be released together with Polyaudio-benchmark, but I don’t see any words in this direction. Do the authors plan to release the datasets?
- Insufficient related work discussion on Large Audio Language Models (section 2.1). While the authors touch upon some LALMs in the introduction, in section 2.1, which should elaborate more on this topic, only includes two references.

**Questions:**

**Questions**

How does the model perform on the rest of the tasks of MMAU Pro? Does it retain good performance albeit the massive training on multi-clips reasoning? A good model should be able to deal with both multi-clips but also single clips, the latter being the most common case in practice.

**Typos/Issues/Suggestions:**

- Line 50: question mark —> missing reference.
- Lines 92-94: Missing comma after “POLYAUDIO-PREF”. Extra full stop at the end of the sentence.
- I suggest that the authors add the citation to each method/benchmark on Table 1, first column, rather than only including the name of them.
- Line 315: a question mark —> wrong Appendix reference.

---

### Official Review · Reviewer_ZjPX · 2025-10-29

**Soundness:** 3
**Presentation:** 2
**Contribution:** 3
**Rating:** 6
**Confidence:** 3

**Summary:**

This paper tackles the challenge that Large Audio Language Models (LALMs) struggle with reasoning across multiple audio clips. It introduces POLYAUDIO, a new model, and POLYAUDIO-BENCH, a new benchmark with 11 tasks, to address this gap. The model, fine-tuned from Qwen2-Audio using GRPO , significantly outperforms baselines on multi-audio tasks while maintaining its original single-audio performance.

**Strengths:**

1. The paper tackles a critical and previously under-addressed gap in LALM capabilities—reasoning across multiple (N-way) discrete audio clips. It moves beyond prior work, which was largely limited to single or pairwise (N=2) audio, by formalizing and evaluating complex reasoning for up to 5 clips.
2. The introduction of POLYAUDIO-BENCH is a significant contribution, providing a comprehensive suite of eleven distinct tasks (detailed in Table 2). This benchmark systematically evaluates a wide range of reasoning skills, from perceptual (e.g., Counting across clips) to compositional (e.g., Factual Consistency), offering a valuable new tool for the research community.

**Weaknesses:**

1. The model's training relies heavily on synthetically generated content, including text from GPT-5 and audio from a TTS model (Higgs Audio v2). The authors acknowledge this limitation , which raises concerns about how well the model will generalize to the acoustic properties, noise, and linguistic diversity of real-world audio.
2. The baseline models, such as Gemini 2.5 Pro and Audio Flamingo 3, do not natively support discrete multi-audio inputs. They were evaluated by concatenating audio clips with silent separators. This is an architectural mismatch that puts them at a fundamental disadvantage, making it difficult to isolate the benefit of POLYAUDIO's training method versus its (more suitable) base architecture.

**Questions:**

1. The benchmark and model are designed for reasoning across "up-to 5" clips. How does the model's performance scale as the number of audio clips increases? For example, how would performance degrade on these 11 tasks if presented with 10 or 20 clips?
2. How would POLYAUDIO perform on a benchmark composed entirely of "in-the-wild" audio? How well can it handle real-world challenges like overlapping speakers, diverse accents, and ambient noise, which are not present in the clean, synthetic training data?

---

### Official Review · Reviewer_9qpU · 2025-10-31

**Soundness:** 2
**Presentation:** 1
**Contribution:** 1
**Rating:** 2
**Confidence:** 4

**Summary:**

The authors fine-tune Qwen2-Audio to tackle the "reasoning over multiple audio clips" task, achieving SOTA results for their proposed PolyAudio-Bench and the multi-audio subset of MMAU-Pro.

**Strengths:**

- Interesting idea of exploring the multi-audio setup for LALM.
- The results for the proposed multi-audio setup look good compared to current SOTA.

**Weaknesses:**

The paper is really hard to follow overall. It seems to be a mix of bad presentation (writing, text structuring) and lack of methodological contributions (e.g., the methodology section is basically non-existent).

The paper's claims are that their contributions rely on a "new Large Audio Language Model" and a new dataset family (PolyAudio-{Train, Pref, Bench}). Neither of them is properly explained in the text.

A few points I collected:
1. The authors multiple times in the text use the term "novel" or "new large audio language model," which is a bit of an overstatement for a fine-tuned version of Qwen2-Audio.
2. The text is missing critical methodology details:
- How the training was done
- Selection of hyperparameters
- How the dataset is composed
- What decisions were made to ensure the quality of the dataset
- What is the difference between the selection criteria of the three different sub-datasets that you compose?
- "We create question–answer pairs using GPT-5 and GPT-OSS120B (OpenAI, 2025)." Which model was used for which questions? Why using a combination of these two models?
3. There are a few references missing (citation appears as ? in the text)
4. Lack of any ablation study
5. Missed opportunity of exploring the performance degradation of curreny LALMs as the number of input audio increases.

**Questions:**

Please address the points in the Weaknesses section above.

---

### Official Review · Reviewer_Vred · 2025-11-01

**Soundness:** 2
**Presentation:** 3
**Contribution:** 1
**Rating:** 2
**Confidence:** 4

**Summary:**

The paper introduces PolyAudio and extends existing large audio language models from single audio to multiple audio by using GRPO. The paper introduces two datasets for training and evaluating the proposed model. The paper achieves strong performance on the multi-audio benchmark.

**Strengths:**

The proposed method is straightforward and achieves good performance on multi-audio reasoning tasks.

**Weaknesses:**

The novelty and the experimental section are very limited. The task-specific synthetic data generation, followed by GRPO fine-tuning pipeline, has been used before to improve performance. Prior works such as R1-Qwen and R1-Omni already demonstrate GRPO + Qwen2audio can achieve SOTA performance on MMAU/MMAR benchmarks.  The proposed method is only evaluated on two datasets, one of which is proposed by the authors. Also, the experimental results for single audio are very limited. PolyAudio is only compared with Qwen2-Audio-7B (Table 7). How does PolyAudio compare to other state-of-the-art models, such as Audio Flamingo 3?

There are several small typos in the paper. For example, lines 50 and 314 have ? for missing references. Line 258 has a missing space between Qwen2-Audio-7B-InstructChu et al.

**Questions:**

How does PolyAudio compare to a simple baseline where the output of each audio is merged via an LLM to produce a unified output?

Does the model generalize across the number of audios? It’ll be interesting to see the impact of the number of audios during training on the multi-audio inference. For example, varying the number of audios during training from 2 to 5 and keeping the number of audios during test as 5.

Does the number of test audios impact the performance?

Does the order of audios impact the performance? For example, do audio1,audio2,audio3 and audio3,audio1,audio2 have similar performance?

What is the impact on performance on different tasks outside of MMAU, such as ASR or speech-to-speech translation?

---

### Note · Authors · 2025-11-12

I have read and agree with the venue's withdrawal policy on behalf of myself and my co-authors.